# BMJ Open Socioeconomic inequalities in dental caries and their determinants in adolescents in New Delhi, India

Manu Raj Mathur,[1] Georgios Tsakos,[2] Christopher Millett,[3] Monika Arora,[1] Richard Watt[2]

## ABSTRACT

**Objectives:** To determine whether socioeconomic inequalities are correlated to dental caries experience and decayed teeth of Indian adolescents, and assess whether behavioural and psychosocial factors mediate this association.

**Methods:** Cross-sectional study of 1386 adolescents living in three diverse areas of New Delhi. Caries experience and number of decayed teeth were assessed clinically and a questionnaire was used to gather sociodemographic and psychosocial data. Zero Inflated Negative Binomial regression models were used to assess the relationship between the outcomes (caries experience and decayed teeth) and area of residence, adjusting for covariates.

**Results:** Significant inequalities in caries experience and number of decayed teeth were observed. Odds of an adolescent being caries free decreased by 66% (OR 0.34, 95% CI 0.23 to 0.49) and 70% (OR 0.30, 95% CI 0.21 to 0.43) in adolescents living in resettlement communities or urban slums, respectively, when compared with the middle class group. No difference was observed among those with caries experience/ decayed teeth. Adjusting for covariates did not affect the inequalities.

**Conclusions:** Area of residence appears to be a very strong and significant determinant for an adolescent to be caries/decay free in India. Psychosocial and behavioural factors do not mediate the association between area of residence and oral health.

## Strengths and limitations of this study

- Very few studies have examined the impact of absolute poverty on the oral health of adolescents in low-income and middle-income countries. This study addresses this gap.
- Fairly large sample size with high response rate.
- Wide array of material, psychosocial and behavioural variables tested through standardised and reliable questionnaire.
- This is a cross-sectional study where data was collected at one point of time.
- Risk of reporting and interviewer bias.
- Measurement issues with the scales used for psychosocial variables and for studying material deprivation.

gradients in oral health have been conducted in high-income countries with populations that generally lie above the poverty line. As such, they do not focus on whether social health inequalities exist in the context of absolute poverty. No study on oral health inequalities from India has considered populations from extremely deprived areas like urban slums and resettlement communities.

Different theories have highlighted various explanations of inequalities observed in general as well as oral health.[4–8] According to these, inequalities arise because of adverse material circumstances, health-affecting behaviours or due to various psychosocial factors. Although there is a considerable amount of literature on general health,[9–11] there is a paucity of evidence in the dental literature for examining how different behavioural, psychosocial and socioenvironmental factors influence oral health inequalities.

Our study assessed the impact of socioeconomic inequalities on dental caries among adolescents living in different geographical areas and conditions in the city of New Delhi, India. We also explored the effect of material, psychosocial and behavioural determinants on these inequalities in dental caries among adolescents.

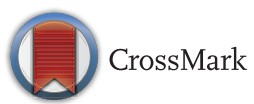

[1]Public Health Foundation of India, New Delhi, India
[2]Department of Epidemiology and Public Health, University College London, London, UK
[3]School of Public Health, Imperial College London, London, UK

**Correspondence to**
Dr Manu Raj Mathur;
manu.mathur@phfi.org

## INTRODUCTION

Oral diseases are among the most common chronic diseases worldwide.[1] Oral diseases not only have an impact on general health and quality of life but may also increase the risk of mortality.[2] Treatment of oral diseases are costly in the healthcare system and for individuals, especially for those from low-income and deprived households.[2]

There are widespread inequalities in oral health outcomes within and between different countries of the world.[3] However, most studies examining social inequalities and

## METHODS

The study was carried out in the National Capital Territory (NCT) of Delhi. Nearly 0.2 million people migrate to Delhi every year and the majority of them reside in urban slums; they constitute about 20% of the total population of Delhi.[12] Many migrants as well as the urban poor also reside in unauthorised and resettlement communities (settlements which have recently been legalised by the Government and were previously slums; these are better off economically in comparison to slums).

### Study population

This cross-sectional study was conducted among adolescents, aged 12–15 years, living in three diverse residential areas of New Delhi reflecting their economic position: urban slums; resettlement communities; and middle and upper middle class communities.

### Study tools

Data were collected through an interviewer-administered questionnaire and a clinical examination. The questionnaire measured material resources, neighbourhood social capital, social support, health-related behaviours (alcohol and tobacco use, diet, frequency of tooth brushing) and key sociodemographic variables. The questionnaire included pre-existing questions and scales which were checked for reliability and validity in the study population during a pilot study. Material resources were assessed using the National Family Health Survey (NFHS) Standard of Living Index which has been validated and used extensively to assess material deprivation in India. Social Capital was measured by a scale developed by Gage et al.[13] This scale has been adapted from Health Behaviour in School Children (HBSC) study conducted by the WHO.[14] Social Support was measured by using Social Support Scale for Adolescents developed by Seidman et al.[15] The questions to assess health-related behaviours in adolescents were derived from the WHO HBSC survey.[14]

A non-invasive clinical examination was performed. We used the Decayed Missing and Filled Teeth (DMFT) index to measure the level of dental caries and decayed teeth.[16] The DMFT index was calculated on every adolescent by using a mouth mirror and a blunt probe. A systematic and standardised approach was used to examine the teeth based on the WHO criteria.[16] Two trained dentists, including the lead investigator, performed the non-invasive clinical dental examination. Examiners were mixed periodically so that no particular examiner was confined to just one particular area for data collection and both examiners were exposed to the broad population. Interexaminer and intraexaminer reliability was checked by repeating the dental examinations on 70 adolescents (5% of the sample). Cohen's unweighted κ coefficient of agreement was used to check for internal consistency. Interexaminer and intraexaminer agreement was above 0.83 for all teeth in the DMFT index.

### Study sample

Slums and resettlement communities were identified from an official list of registered resettlement communities and urban slums. The inclusion criteria were (A) communities within a radius of 25 km from the research office, (B) slum and resettlement community present together as a cluster, (C) more than 500 households in each component of the cluster and (D) have a known non-governmental organisation working for the community and willing to participate in the research. We identified 14 slums and resettlement communities. A census was performed in each of these communities to collect demographic data.

Adolescents from middle and upper middle class households in India generally study in private schools which have English as the medium of education and charge higher fees ('English Medium Schools'). These schools were targeted to obtain the desired sample of adolescents belonging to middle and upper middle class homes. Inclusion criteria for English medium schools were: (A) those having secondary level classes, (B) present in the same vicinity as that of the low-income communities in the sampling frame, (C) having at least 40 pupils per class and (D) being coeducational (boys and girls).

We used multistage random sampling. Five slums and resettlement communities were randomly selected from the 14 identified communities. Among the selected communities, all households having at least one child in the age group of 12–15 years were identified. Eligible households were randomly selected and approached to request adolescents to participate in the study. If a household had two or more eligible adolescents, then all of them were invited to participate. Five English medium private schools were randomly selected from a list of 48 identified schools. From the selected schools, grades which normally have children of 12–15 years of age were identified. The class teacher then assigned random numbers to eligible adolescents in order to maintain the anonymity of the identity of students. Adolescents were then randomly selected from the provided list of numbers. Both the parents and adolescents signed the consent forms after having been informed about the study.

Sample size was calculated for difference in means of the two clinical outcomes (caries experience and decayed teeth) between the three different adolescent groups, with 80% power and 5% significance level. The differences in mean values of the clinical outcomes were obtained through a pilot study conducted on 150 adolescents from a setting similar to that of this study. The calculated sample size was increased by a factor of 25% to account for potential non-response, and then by a factor of 1.3% to account for the effect of clustering. The estimated sample size was 1338 adolescents (446 adolescents per group).

### Variables

Main explanatory variable in this study was the adolescent's socioeconomic position assessed through area of

**Table 1** Socioeconomic inequalities in caries experience and decayed teeth

| Variable | DMFT | | Carious teeth | |
|---|---|---|---|---|
| | Mean (95% CI) | Median* (IQR) | Mean (95% CI) | Median* (IQR) |
| Middle/upper middle class | 0.96 (0.82 to 1.21) | 0 (0–1) | 0.72 (0.59 to 0.85) | 0 (0–1) |
| Resettlement communities | 1.38 (1.23 to 1.54) | 1 (0–2) | 1.34 (1.19 to 1.50) | 1 (0–2) |
| Slums | 1.74 (1.55 to 1.93) | 1 (0–3) | 1.58 (1.40 to 1.76) | 1 (0–2) |

*p<0.001.
DMFT, Decayed Missing and Filled Teeth.

residence (slums, resettlement communities, middle and upper middle class homes). Covariates are grouped into the following categories: (A) material resources, (B) neighbourhood social capital (bridging and bonding types of social capital measuring trust, norms and reciprocity in a community), (C) social support and (D) health-related behaviours (diet, tobacco and alcohol use, brushing frequency, visit to a dentist, getting bullied and involvement in physical fight). All these covariates were significantly associated with socioeconomic position in bivariate analyses (results not shown) and were accounted for in the multivariable models. Outcome variables were dental caries experience and prevalence of decayed teeth.

### Data analysis
Descriptive statistics were calculated to assess the frequency distributions of explanatory and outcome variables. As there were a high number of zeros (caries-free/decay-free teeth) in the outcomes and the variance was considerably greater than the mean, Zero Inflated Negative Binomial (ZINB) regression analysis was used to assess the association between area of residence and caries experience (DMFT) and the number of carious teeth (decayed teeth; DT).[17] A ZINB regression generates two separate models. The first model is a logit model generated for a 'certain zero' in the outcome (ie, if no decayed, missing or filled teeth if DMFT or no active caries if DT is the outcome) and predicts whether or not an adolescent would be in this group. The second model is the negative binomial model predicting the count of those adolescents who are not in the 'certain zero group' ,that is, by checking the number of decayed, missing or filled teeth among those with DMFT>0 or at the number

of decayed teeth among those with DT>0, respectively. All analyses were conducted in Stata V.12.

### RESULTS
A response rate of 86.6% was achieved (n=1386). There were 736 (53.1%) boys and 650 (46.9%) girls; proportions that are almost the same to the gender distribution of Delhi NCT (53.6% males and 46.4% females).[18] Overall, 460 (33.2%) adolescents belonged to the middle and upper middle class group, 462 (33.3%) were from resettlement communities and 464 (33.5%) from urban slums. Almost half (49.7%) of the clinically examined adolescents had previous caries experience. The mean DMFT was 1.36 (1.27 to 1.46). Of the 689 adolescents with caries experience, 644 had decayed teeth at the time of clinical examination (mean=1.21; 95% CI 1.12 to 1.31).

There was a clear social gradient, with consistently greater levels of caries experience (DMFT) at each lower level of area of residence of adolescents (p<0.0001). Adolescents from middle/upper middle class homes had mean DMFT of 0.96 (95% CI 0.82 to 1.21), those from resettlement communities had a mean of 1.38 (95% CI 1.23 to 1.54) and those from urban slums had a mean DMFT of 1.74 (95% CI 1.55 to 1.93). Similarly, the mean number of decayed teeth was higher at each lower socioeconomic group (p<0.0001). The mean number of decayed teeth in adolescents from middle/upper middle class homes was 0.72 (0.59 to 0.85), in those from resettlement communities 1.34 (1.19 to 1.50) and in those from urban slums 1.58 (1.40 to 1.76; table 1 and figure 1).

Table 2 shows the results of the ZINB regression models for caries experience. Adjustment for covariates

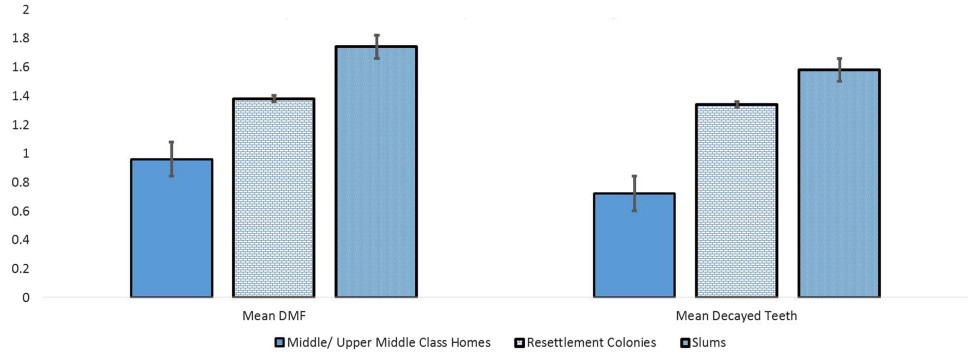

**Figure 1** Gradient in caries experience and mean decayed teeth according to area of residence.

**Table 2** Association between area of residence and caries experience adjusting for demographic variables, health-related behaviours, material resources, social support and social capital† (N=1386)

| | Area of residence | | |
| --- | --- | --- | --- |
| | Middle class OR (95% CI) | Resettlement community | Slums |
| Logit | | | |
| Unadjusted | 1 | 0.34 (0.23 to 0.49)* | 0.30 (0.21 to 0.43)* |
| Adjusted for age, sex and religion | 1 | 0.28 (0.19 to 0.41)* | 0.25 (0.17 to 0.37)* |
| Fully adjusted for all covariates‡ | 1 | 0.22 (0.12 to 0.39)* | 0.22 (0.11 to 0.46)* |
| | | IRR (95% CI) | |
| Negative binomial | | | |
| Unadjusted | 1 | 0.88 (0.73 to 1.05) | 1.06 (0.89 to 1.26) |
| Adjusted for age, sex and religion | 1 | 0.91 (0.76 to 1.09) | 1.13 (0.94 to 1.35) |
| Fully adjusted for all covariates‡ | 1 | 1.00 (0.76 to 1.34) | 1.10 (0.78 to 1.54) |

*p<0.001.
†Association tested using Zero Inflated Negative Binomial Regression Method.
‡Fully adjusted for age, sex, religion, health-related behaviours, material resources, social support and social capital.
IRR, Incidence Rate Ratio.

did not have a considerable effect on the inequalities identified. Compared with the middle/upper middle class adolescents, those living in resettlement communities had a significantly lower OR of being caries free (OR=0.33; 95% CI 0.23 to 0.49 in the unadjusted model; OR=0.22; 95% CI 0.12 to 0.39 in the fully adjusted model), and the same was the case for those living in urban slums (OR=0.30, 95% CI 0.21 to 0.43 in the unadjusted model and OR=0.22, 95% CI 0.11 to 0.46 in the fully adjusted models). In contrast, there were no differences between the three residential sites in relation to the number of teeth with caries experience (DMFT >0).

The respective results for decayed teeth were similar, with a significantly lower OR of being decay free for adolescents from resettlement communities (OR=0.25; 95% CI 0.17 to 0.37) and for those from urban slums (OR=0.24; 95% CI 0.16 to 0.35) compared with their more affluent counterparts. The fully adjusted ORs were 0.21 (95% CI 0.14 to 0.32) for adolescents from resettlement communities and urban slums in comparison with adolescents from middle/upper middle class homes. There were no significant differences between the three groups in the number of adolescents who had experienced decayed teeth as compared with those currently having decayed teeth (table 3).

## DISCUSSION

We observed a monotonic gradient for the differences in caries experience and decayed teeth between adolescents living in diverse residential areas of New Delhi. Our results showed that there is a significant difference between the proportion of individuals who were caries free or individuals with decay-free teeth between the three areas of residence. However, once an individual experienced caries or developed tooth decay, it did not matter as to which residential group they belonged to as there were no significant differences in the probability of having one or more carious or decayed teeth between the three groups.

The study population comprised of adolescents from areas with extreme deprivation, such as urban slums. Only two studies have looked at the influence of socioeconomic inequalities on dental caries in India,[19 20] but these did not study adolescents from extremely deprived areas. Our findings in relation to the association of socioeconomic inequalities with caries experience and number of decayed teeth are similar to previous studies conducted on adolescents[21–25] and children.[26–29]

We used area of residence as an indicator of the socioeconomic position of the adolescents. Only one previous study has looked at inequalities in dental caries by using an area-based measure of socioeconomic position.[27] Thomson and Mackay (2004), in their study on 9-year-old school children from New Zealand, used area-based as well as individual-based measures of socioeconomic position and found that the inequalities in adolescent dental caries were steeper when area-based measures were used to define the socioeconomic position.

Adjusting for all others factors simultaneously in our study did not cause a noteworthy change in the associations, showing that the combination of different factors investigated in this study had a limited effect on the observed inequalities in the incidence of adolescent dental caries. Jung et al (2011) showed that behavioural (brushing frequency, diet, smoking and alcohol use) and family affluence had no influence on the socioeconomic inequalities observed for self-reported toothache, bad breath and fractured teeth among South Korean adolescents. However, when combined with psychosocial factors (perceived stress and happiness), these factors partially accounted for the inequalities seen in oral health.[30]

We collected data on oral hygiene-related (tooth brushing frequency, dental visit frequency) and health-affecting (tobacco, alcohol, involvement in physical fight, diet) behaviours to understand the effect of these behaviours on socioeconomic inequalities in oral health. None of the

**Table 3** Association between area of residence and decayed teeth adjusting for demographic variables, health-related behaviours, material resources, social support and social capital† (N=1386)

| | Area of residence | | |
| --- | --- | --- | --- |
| | Middle class OR (95% CI) | Resettlement community | Slums |
| **Logit** | | | |
| Unadjusted | 1 | 0.25 (0.17 to 0.37)* | 0.24 (0.16 to 0.35)* |
| Adjusted for age, sex and religion | 1 | 0.21 (0.14 to 0.31)* | 0.20 (0.13 to 0.30)* |
| Fully adjusted for all covariates‡ | 1 | 0.21 (0.14 to 0.32)* | 0.21 (0.14 to 0.32)* |
| | | IRR (95% CI) | |
| **Negative binomial** | | | |
| Unadjusted | 1 | 0.95 (0.78 to 1.17) | 1.11 (0.91 to 1.35) |
| Adjusted for age, sex and eeligion | 1 | 1.02 (0.83 to 1.25) | 1.21 (0.99 to 1.47) |
| Fully adjusted for all covariates‡ | 1 | 1.01 (0.82 to 1.24) | 1.19 (0.97 to 1.46) |

*p<0.001.
†Association tested using Zero Inflated Negative Binomial Regression Method.
‡Fully adjusted for age, sex, religion, health-related behaviours, material resources, social support and social capital.
IRR, Incidence Rate Ratio.

health-affecting behaviours had any significant effect on inequalities in oral health observed in our study. Most of the studies on adolescents and young children have also shown a negligible or a minor effect of health-related behaviours on inequalities in oral health.[24 31 32]

Material deprivation in our study was measured through the NFHS standard of living index.[33] This index was first developed in 2000 and assessed the availability of basic material things required for living by an individual. However, India has since seen rapid economic development leading to a general improvement in the standard of living. Therefore, some of the items and respective weights used in the standard of living index may not be equally relevant in the current situation. This measurement issue may partly explain why we were not able to see any effect of material deprivation on inequalities in incidence of dental caries.

Social capital is a multidimensional concept described by different authors in different ways and therefore, is not easily measured with only a few items.[34] Putnam[35] in his description of social capital stressed that community participation is also an intertwined feature along with trust and norms of reciprocity, and forms an important component of social capital. The social capital questionnaire[13] used in our study measured the trust and norms of reciprocity in the society but did not measure the level of community participation, which might be one of the reasons of not finding any effect of social capital on inequalities in caries experience and decayed teeth. While social support may be seen as bi-directional (receiving as well as giving),[36] our scale of social support[15] measured mainly the received support or support available to an adolescent and did not assess the aspect of 'giving' support to others.

Adolescents were sampled from extremely deprived urban slums and deprived resettlement areas of New Delhi thus providing a realistic reflection of oral health inequalities in urban areas. We adopted scales and questions from internationally validated questionnaires, and further tested and adapted these for use on Indian adolescent populations. We acknowledge a number of study limitations in addition to the measurement issues about material deprivation and psychosocial variables described above. We studied only social capital and social support from the vast array of psychosocial variables. There are many other psychosocial variables like stress, depression and anxiety which were not investigated. Thus, the results do not fully cover the entire psychosocial pathway to oral health inequalities. Furthermore, reporting bias cannot be ruled out as adolescents who were well versed with the consent form and objectives of the study might have given responses that are either socially desirable or perceived to be 'wanted' by the interviewer.

Our study findings suggest that the relative impact of deprivation on oral health inequalities is seen only in individuals who are disease free, with a clear gradient indicating higher prevalence of adolescents free from caries (or caries experience) for each consecutively less deprived area of residence. Our study has also shown that area of residence may be a very important determinant of the oral health status of adolescents in India. Psychosocial, material or behavioural characteristics did not mediate the role of extreme living conditions on oral health. This finding highlights the importance of health promotion[37] in reducing inequalities in oral health. In order to reduce inequalities in dental caries experience, there is a need to intervene early and prevent the onset of dental caries and 'act before it happens' rather than intervening after caries has affected the population. There is a need to design policies which aim at primary prevention and improving health by taking action on the broader structural determinants of oral health.

**Contributors** MRM, RW, GT and MA conceptualised and designed the study. MRM collected the data, performed analysis and wrote the manuscript. RW and GT directed the development of methodology, analytical plan and interpretation of results. MRM, RW, GT and CM undertook the critical revisions of the paper for substantial intellectual content. MA contributed to the background and methods section of this paper. All the authors approved of the final version to be published.

**Funding** This work was supported by a Wellcome Trust Capacity Strengthening Strategic Award to the Public Health Foundation of India and a consortium of UK universities. CM is funded by a NIHR Research Professorship award.

**Competing interests** None.

**Ethics approval** University College London Research Ethics Committee and Public Health Foundation of India Technical Review and Institutional Ethics Committee.

**Provenance and peer review** Not commissioned; externally peer reviewed.

**Data sharing statement** No additional data are available.

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
