## [Reviewer comments · BMJ Open]

Some articles will have been accepted based in part or entirely on reviews undertaken for other BMJ Group journals. These will be reproduced where possible.

ARTICLE DETAILS

TITLE (PROVISIONAL)	Socio-economic inequalities in dental caries and their determinants in adolescents in New Delhi, India
AUTHORS	Mathur, Manu; Tsakos, Georgios; Millett, Christopher; Arora, Monika; Watt, Richard

VERSION 1 – REVIEW

REVIEWER	Professor Lilani Ekanayke Deaprtment of Community Dental Health Faculty of Dental Sciences university of Peradeniya oeradeniya Sri Lanka
REVIEW RETURNED	21-Oct-2014

GENERAL COMMENTS	The manuscript assessed the socio-economic inequalities in caries experience and decayed teeth of Indian adolescents and the effect of behavioural and psychosocial factors on these inequalities. It is a well designed study as well as the manuscript is well written. However the reviewer has a few issues which need to be addressed prior to considering it for publication. Abstract: Conclusions should summarize the findings adequately in line with the objectives. However the findings related to the second objective are not given.Methods: what about inter/intra variability associated with caries diagnosis?How many slums/resettlement communities were selected from a total of 14?How many schools were selected from a total of how many?Inclusion criteria are given with respect to selection of communities/schools? But what about inclusion/exclusion criteria with regards to selection of subjects? They need to be given.It is stated that the sample size was calculated for differences in means of the two clinical outcomes. From where did the researchers obtain these data?What was the statistical software used for data analysis? This should be mentioned.Did the researchers obtain informed consent from
--

	parents/children? Nothing is mentioned about this 9. Results: the researchers have obtained information pertaining to several material, psychosocial and behavioural variables. However no data pertaining to these are given. Were these variables significantly associated with socio-economic status? Were they also socially patterned? If they were not significantly associated with SE status then they cannot be considered as possible determinants. It is necessary and important to give the results of a bivariate analysis of these variables with socio-economic status as assessed by place of residence in a table. 10. Table 1: What test was used to compare data in the table? As the data are presented in terms of the “mean”, I presume it is “Anova”! As both DMFT and decayed teeth were not “normally distributed”, to compare groups a non-parametric test should be used i.e -K wallis test and the data in Table 1 should be presented in terms of the median and range rather than the mean. 11. Tables 2 and 3: footers indicate a single #, but the tables do not show this. 12. Discussion: page 16 line 47 “the relative effects of deprivation on oral health inequalities is seen only in individuals who are disease free”. I am a bit concerned about this statement. Suggest rephrasing it to indicate that those who were caries free were higher in the high SE groups compared to the others.
--	---

REVIEWER	John Skinner Centre for Oral Health Strategy, New South Wales, Australia
REVIEW RETURNED	24-Oct-2014

GENERAL COMMENTS	This is an important study and overall a well written paper. Just minor revision of grammar in several places will improve readability of the paper and clarity of the findings.
--

VERSION 1 – AUTHOR RESPONSE

Reviewer 1

Comment 1:

Please add New Delhi to the title as this is not an India-wide study

Response:

The title has been amended and New Delhi added in the title.

Comment 2:

Abstract: Conclusions should summarize the findings adequately in line with the objectives. However the findings related to the second objective are not given.

Response:

We have added the findings related to second objective in the abstract text.

Comment 3:

Methods: what about inter/intra variability associated with caries diagnosis?

Response:

Two trained dentists including the lead investigator performed the non-invasive clinical dental examination. They were mixed periodically so that no examiner was confined to just one particular area for data collection and both examiners were exposed to all three different population strata sampled in this study. Inter- and intra-examiner reliability was checked for dental caries by repeating examinations on 70 adolescents (5% of the sample). Cohen's Unweighted Kappa coefficients indicated high levels of both inter- and intra-examiner agreement ($K > 0.83$ in all cases).

The necessary details are included in the main text (Page 7).

Comment 4:

How many slums/resettlement communities were selected from a total of 14?

Response:

Five slums and resettlement communities were randomly selected from the fourteen identified low income communities. The necessary text has been added in the main document (Page 8).

Comment 5:

How many schools were selected from a total of how many?

Response:

Five English medium private schools were randomly selected from a list 48 identified schools which matched the inclusion criteria.

Comment 6:

Inclusion criteria are given with respect to selection of communities/schools? But what about inclusion/exclusion criteria with regards to selection of subjects? They need to be given.

Response:

All adolescents that were 12-15 years of age in the selected schools were eligible to participate in the study and a random sample was drawn. The only other criterion for recruitment was that both the parents and adolescents had understood the objectives of the study and signed the informed consent forms. This has been included in the main text (Page 8).

Comment 7:

It is stated that the sample size was calculated for differences in means of the two clinical outcomes. From where did the researchers obtain these data?

Response:

The differences in mean values of the clinical outcomes were obtained through a pilot study conducted on 150 adolescents from a setting similar to that of this study. The calculated sample size was increased by a factor of 25% to account for potential non-response, and then by a factor of 1.3 to account for the effect of clustering. The estimated sample size was 1338 adolescents (446 adolescents per group).

These details are added in the main text (Page 8).

Comment 8:

What was the statistical software used for data analysis? This should be mentioned.

Response:

All analyses were conducted in Stata 12. This has been added to the main text (Page 10).

Comment 9:

Did the researchers obtain informed consent from parents/children?

Response:

Before commencing with the data collection informed consent was obtained from the parents/guardians of the adolescents as well as the adolescents themselves. Information sheets were prepared to inform participants of the details of the study. All data collected followed standard data Protection policies.

The relevant details have been included in the main text (Page 8).

Comment 10:

Results: the researchers have obtained information pertaining to several material, psychosocial and behavioural variables. However no data pertaining to these are given. Were these variables significantly associated with socio-economic status? Were they also socially patterned? If they were not significantly associated with SE status then they cannot be considered as possible determinants. It is necessary and important to give the results of a bivariate analysis of these variables with socio-economic status as assessed by place of residence in a table.

Response:

The evidence of the association of various behavioural, psychosocial and material factors with Socioeconomic position (SEP) is well established in the literature. We can also confirm that bivariate associations between these variables and SEP were statistically significant ($p < 0.0001$ for all the variables). We acknowledge the point raised by the reviewer and now mention this finding in the text (page 9). However, we do not feel it necessary to present these data in an Appendix of the paper but would be happy to do so if this is editorial preference.

Comment 11:

Table 1: What test was used to compare data in the table? As the data are presented in terms of the "mean", I presume it is "Anova"! As both DMFT and decayed teeth were not "normally distributed", to compare groups a non-parametric test should be used i.e -K wallis test and the data in Table 1 should be presented in terms of the median and range rather than the mean.

Response:

Kruskal-Wallis test was used to compare the differences in DMFT and number of decayed teeth between the three study groups as the data was not normally distributed. We presented the mean values in the Table because this is the conventional way of depicting the DMF and D scores in the dental literature. Therefore we used it in order to allow comparability with other studies. We agree with the reviewer that since the data is not normally distributed, we should also report median values and inter-quartile range and have made the necessary amendments in Table 1.

Comment 12:

Tables 2 and 3: footers indicate a single #, but the tables do not show this.

Response:

The necessary correction has been made.

Comment 13:

Discussion: page 16 line 47 "the relative effects of deprivation on oral health inequalities is seen only in individuals who are disease free". I am a bit concerned about this statement. Suggest rephrasing it to indicate that those who were caries free were higher in the high SE groups compared to the others.

Response:

The usefulness of ZINB is that we can model the two distinct parts of the distribution (0s and non-0 count) separately. We have shown that the gradients exist for the 0s (i.e. Caries experience free) but not for those that did not score 0 on the outcome. We feel this is a unique contribution of our study. We have changed the wording of this sentence to better clarify this: " the relative impact of deprivation on oral health inequalities was seen only in individuals who were disease free, with a clear gradient indicating higher prevalence of adolescents free from caries (or caries experience) for each consecutively less deprived area of residence" (page 16).

Reviewer: 2

Comment:

This is an important study and overall a well written paper. Just minor revision of grammar in several places will improve readability of the paper and clarity of the findings.

Response:

We thank the Reviewer for this positive comment. Grammar has been corrected at several places indicated in YELLOW colour in the main text to improve readability and for clarity of findings.

We hope that you would find these changes acceptable.

VERSION 2 – REVIEW

REVIEWER	Professor Lilani Ekanayke Department of Community Dental Health Faculty of Dental Sciences University of Peradeniya Peradeniya Sri Lanka
REVIEW RETURNED	26-Nov-2014

GENERAL COMMENTS	The researchers have addressed all issues raised at the first review
--